# Effects of Vaccination against COVID-19 in Chronic Spontaneous and Inducible Urticaria (CSU/CIU) Patients: A Monocentric Study

**DOI:** 10.3390/jcm11071822

**Published:** 2022-03-25

**Authors:** Teresa Grieco, Luca Ambrosio, Federica Trovato, Martina Vitiello, Ilaria Demofonte, Marta Fanto, Giovanni Paolino, Giovanni Pellacani

**Affiliations:** 1Dermatology Unit, Sapienza University of Rome, 00185 Rome, Italy; teresa.grieco@uniroma1.it (T.G.); ambrosioluca1@gmail.com (L.A.); federica.trovato01@libero.it (F.T.); martinavitiello94@gmail.com (M.V.); ilaria.demofonte@gmail.com (I.D.); martafanto@gmail.com (M.F.); giovanni.pellacani@uniroma1.it (G.P.); 2Unità di Dermatologia, Ospedale San Raffaele, 20121 Milano, Italy; 3Dermatologia Clinica, Università Vita-Salute San Raffaele, 20121 Milano, Italy

**Keywords:** urticaria, chronic, COVID-19, vaccines, SARS-CoV-2, cutaneous, skin, reactions

## Abstract

Background: Patients affected by pre-existing chronic spontaneous/Inducible urticaria (CSU/CIU) still feel unsafe due to the potential risk of an Adverse Event Following Immunization (AEFI) and Cutaneous Adverse Reactions (CARs) of COVID-19 vaccines. The appropriate management in this field remains debated and evidence is still lacking. Methods: We considered 160 CSU/CIU patients in Omalizumab/antihistamine therapy who received two doses of Comirnaty/Moderna mRNA vaccines; 20 of them also received a booster dose. Urticaria Activity Score-7 (UAS7) was used to assess the severity of the disease. Demographics, medical history, AEFI and CARs outcome after vaccination were collected by administering a web-based questionnaire completed by phone interview. Results: In total, 147 patients did not show urticaria relapse (91.88%). Worsening cutaneous symptoms were experienced by 13 of our patients (8.12%). Exacerbation had a mean duration of 2 days and 11 h and mostly occurred after the first dose (69.23%). Systemic mild side effects were experienced by 9 patients (5.62%). No severe reactions were observed. Conclusions: Omalizumab can potentially prevent CARs and AEFI; however, major problems were registered during the 2-month stop period scheduled in the treatment. We suggest patients should not undergo vaccination during this period. CSU/CIU exacerbations appear to be transient and can be managed by antihistamines.

## 1. Introduction

Chronic Spontaneous Urticaria is a relapsing mucocutaneous disease, characterized by hives (40%) angioedema (20%) or both (40%) without any apparent cause. It affects more than 50 million people globally [1]; females have twice the risk of developing CSU than males and onset is most common between 30 and 40 years, but it can occur at any age. This condition severely interferes with social habits and impairs the sufferer’s quality of life [2,3,4,5,6,7,8].

In December 2020, a huge vaccination campaign began in Europe with the introduction of anti-COVID-19 vaccines, such as Pfizer mRNABNT162b2 (Comirnaty^®^), COVID-19 Vaccine Moderna mRNA-1273 (Spikevax^®^) and the viral vector-based vaccine Astrazeneca AZD1222 (Vaxvzevira^®^). Among them, Pfizer mRNABNT162b2 (Comirnaty^®^), and COVID-19 Vaccine Moderna mRNA-1273 (Spikevax^®^) are currently the most injected. Both vaccines consist of mRNA strands encoding SARS-CoV-2 Spike glycoprotein (S-protein), delivered to cells by neutrally charged lipid nanoparticles (LNP). The genetic sequence of both vaccines is slightly altered by two proline substitutions (K986P and V987P) and compared to viral vector vaccines they show a better safety profile [8].

The management of CSU includes second-generation H1 antihistamines as first-line treatment, which provide complete control in less than 50% of the patients [5]. In refractory patients, treatment with Omalizumab, a humanized IgG1 recombinant monoclonal antibody targeting IgE antibodies, is recommended as monotherapy, as well as an add-on therapy to H1-antihistamines. Biologic therapy is considered safe and effective in the long-term monitoring of CSU; in fact, very few cases of allergic severe reactions have been reported in the literature [6]. In terms of controlling the disease activity, frequency of relapses and/or longstanding history of chronic urticaria, several potential clinical or serological biomarkers have been suggested: D-dimer, total IgE level and coexistent autoimmunity conditions have been recently suggested and are currently used in monitoring the disease [7].

Since the beginning of the pandemic, the two mRNA COVID-19 vaccines Pfizer mRNABNT162b2 (Comirnaty) and COVID-19 Vaccine Moderna mRNA-1273 (Spikevax) have been in use. mRNA vaccines may induce a transient inflammatory response, possibly triggered by the endocytosis of the mRNA-LNPs complex by natural immunity resident cells (mast cells, eosinophils), after the intramuscular injection. Local and recruited Antigen-Presenting Cells (APCs) express the S-protein mRNA, undergoing activation and maturation. The activated APCs and the circulating mRNA-LNPs and S-proteins brings the S-antigen to both B and T cells; B cells recognize the S-antigen, leading to the generation of neutralizing antibodies against the S-protein [9,10,11].

Published data have shown that both vaccines provide immunity against a SARS-CoV-2 infection, by inducing similar humoral response, without any difference in cellular immunity. There is, however, preliminary unpublished evidence that Comirnaty vaccine may trigger a stronger CD8 T cells-response than Spikevax [12]. Together with a good efficacy to limit Sars-CoV-2 expansion, there is a relative increase in adverse events associated with the use of COVID-19 vaccines. General manifestations classified as Adverse Events Following Injection (AEFI) usually include fever, fatigue, headache, muscle and joint pain, diarrhoea, as well as local injection-site reactions. The majority of these symptoms are reported as mild and transient, occurring mostly after the second dose [12,13]. Cutaneous Adverse Reactions (CARs) are rarely reported and urticaria-angioedema, locally or generally distributed, is the mildest adverse skin event observed in 0.6% of total CARs in a general population [14]. In this spectrum, and according to the literature, type I and type IV hypersensitivity reactions were the most observed. Specifically, referred data suggest urticaria and angioedema were the mainly observed Type-I reactions; moreover, cases of new onset urticaria have been reported by other authors [15,16]. An American registry-based study of 414 cases firstly reported a percentage of 1.7% of patients who developed a cutaneous urticaria rash after Pfizer mRNABNT162b2 (Comirnaty) and Moderna mRNA-1273 (Spikevax) vaccines [17].

Additional manifestations include several cases of generalized morphoea, herpes-like rashes, rosea-like pityriasis rashes, lichenoid rashes and sarcoid-like reaction [18,19,20]. Focusing on urticaria, data on tolerability in patients affected by CSU/CIU in real life are currently lacking; moreover, our aim was to evaluate temporally associated CARs and AEFI observed in our dermatology and allergology outpatient clinic.

## 2. Materials and Methods

During the period from 1 November 2021 to 10 December 2021, 160 patients with a previous diagnosis of CSU/CIU were admitted to the study. The patient sample consisted of 99 females and 61 males (male/female ratio 1:1.62), with a mean age of 48.4 years (range = 18–91). Out of 160 patients, 104 were treated by Omalizumab (65%); fifteen (*n* = 15) patients out of 104 (14.42%) were taking anti-H1 as add-on therapy as needed, while fifty-six (*n* = 56) were treated by anti-H1 (35%) at least twice-a-day.

Urticaria Activity Score-7 (UAS7) was used to assess the severity of the disease and to assess the severity of the exacerbations, by the comparison of UAS7 before the administration of the vaccine and 3 days after the administration (ΔUAS7). The cutaneous condition was well controlled in all our patients (UAS7 < 16): in patients treated with anti-H1 the mean value of UAS7 was 13.7 and in patients treated by Omalizumab the mean value of UAS7 was 2.52. Subjects were fully vaccinated with at least two doses of the vaccine; 20 of them had received the third booster dose.

All patients provided verbal consent and received a link to the online survey; after clicking the link, patients were asked to confirm their intention to proceed willingly. We completed the survey by phone. Patients were clearly informed that no personal data (names, IP or email addresses) unnecessary to the survey would be collected to preserve anonymity. Participation was free and no compensation was offered.

The web-based survey was conducted on the platform Google Forms (Google LLC, Menlo Park, CA, USA). Every patient was administered the same questionnaire formulated in Italian. All answers were self-reported and not externally validated. The questionnaire investigated the following features: demographics including age and sex; clinical features including time of onset of the CSU, personal history related to COVID-19 vaccination, including the status of CSU (relapsing, remitting or fully controlled) at time of vaccination. Patients were also asked about prior COVID-19 infection.

The telephone survey was conducted to give assistance to the older patients who were unable to fill in the form.

## 3. Results

The results of our investigation survey highlighted 13 patients (8.12%) who experienced exacerbation of CSU symptomatology after vaccine administration. Within the study period, in the subgroup of patients who were taking omalizumab, three of them showed relapsing symptoms of urticaria (1.92%); two (66.6%) of them during the stop-period of 8 weeks following 6 months of continuous therapy, as stated by the Italian Medicines Agency. Out of the 56 patients who were taking anti-H1 in monotherapy, 10 of them (17.86%) showed worsening of pre-existing urticaria (average ΔUAS7 = 3.12). Exacerbation occurred in nine patients (69.23%) after Pfizer mRNABNT162b2 (Comirnaty) and in four patients (30.77%) after Moderna mRNA-1273 (Spikevax). In total, 147 (*n* = 147) patients out of 160 did not show any relapse in symptoms of urticaria (92.45%). No case of angioedema was observed. (Figure 1, Figure 2 and Figure 3).

One medium aged patient affected by CSU associated with CIU (Cholinergic Urticaria) presented with a systemic reaction characterized by massive arm oedema, associated with persistent painful consensual axillary adenopathy, generalized urticaria, fever and headache. Symptoms underwent remission in 5 days and were fully responsive to antihistamines and paracetamol. Exacerbation mostly occurred after the first dose (9 patients out of 13–69.23%), three after the second dose (27.08%) and one after the third dose (7.69%). The only patient who experienced a relapse in symptoms following the third dose of the vaccine discontinued Omalizumab during the stop-period.

The exacerbation lasted from 24 h to 5 days, with a mean duration of 2 days and 11 h. It was managed by adding anti-H1 therapy twice-a-day. Out of our sample study-of 160 subjects, 9 patients (5.63%) reported CARs and AEFI mostly after the second dose (7 out of 9–77.77%), with a mean duration of 20 h. AEFI, as fever headache and asthenia, were successfully treated with acetaminophen. No patients had a history of prior COVID-19 infection.

## 4. Discussion

Monitoring patients by questionnaire administration can offer a prompt system to evaluate and prevent any problems occurring in CSU/CIU patients upon follow-up in specialized centres, particularly in the COVID-19 era and related to fear of vaccination; indeed, knowledge about real life data is even more necessary.

In our previous study conducted on a cohort of 2740 subjects admitted to COVID-19 Pfizer and Moderna vaccinations in our hospital, the incidence of urticarial rashes represented 28% of all CARs [14]. Similarly, Robinson et al. described rash and itching as the most common cutaneous reaction in patients who received mRNA COVID-19 vaccines registered after the first dose (1%) and after the second dose (1.6%), respectively [15]. Of our sample study, 147 patients out of 160 did not show any relapse in symptoms of urticaria (92.45%). No severe adverse reaction or angioedema were observed. The most complained symptoms (AEFI) were arm pain (12.5–2.88%), asthenia (8.93–2.88%), headache (7.14–1.92%) and fever (3.57–1.92%). All 10 patients (17.86%) who showed worsening pre-existing urticaria (average ΔUAS7 = 3.12) were undertaking only antihistamines twice a day. The remaining three patients were under omalizumab therapy; one patient developed oedema of arm, generalized flare of urticaria, axillary adenopathy, fever and headache after the booster of Comirnaty mRNA vaccine. The administration was performed during the two-month stop period of Omalizumab. CSU exacerbation mostly occurred after the first dose, specifically in 9 patients (69.23%). Recommendations exist concerning the management of the booster dose in subjects that developed CARs after the previous vaccine administration, but no evidence emerged in our study confirming the caution in completing the vaccination. Cases of worsening or relapse of the disease appear to be transient and were managed by antihistamine therapy twice a day. All adverse reactions resolved in a maximum of 5 days.

As reported by Gambichler et al., type I allergic reactions following COVID-19 vaccinations is mainly due to polyethylene glycole (PEG) and structurally-related polysorbate-80 were considered as potential triggers of both IgE and non-IgE-mediated reactions [19,20,21]. According to recently published data, there is no convincing evidence of the clinical relevance of hypersensitivity reactions in PEG/Polysorbate sensitized patients who received COVID 19-mRNA vaccines [22]. In this regard, according to the German Society of Allergology and Clinical Immunology (DGAKI) and the German Society for Applied Allergology (AeDA) position paper, omalizumab should be continued during the vaccination period, and systemic antihistamines do not impact the vaccination effect [23]. Our data underline that omalizumab can potentially prevent urticaria flares (CARs) and AEFI in patients with CSU, despite some major problems that may occur during the 2-month stop period scheduled in the treatment. Considering our experience, we suggest patients should not undergo vaccination during this period.

## 5. Conclusions

In conclusion, according to our sample, COVID-19 vaccination in the CSU/CIU can be considered safe and is advisable. Cases of exacerbation or worsening of the disease appear to be transient and can be managed by antihistamine therapies. Patients with well-controlled urticaria (assessed by UAS7 < 16) who undergo Omalizumab seem to be more protected against potential urticaria flares and AEFI. In general, for patients with CSU, anti-SARS-CoV-2 vaccines also currently maintain a good general safety profile.

## Figures and Tables

**Figure 1 jcm-11-01822-f001:**
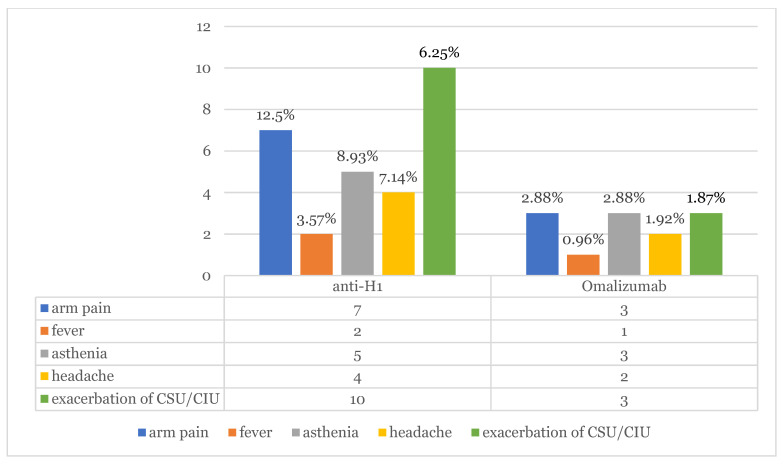
Adverse Events Following Immunization (AEFI) and exacerbation of CSU/CIU symptoms observed in patients taking antihistamines and omalizumab therapy, respectively.

**Figure 2 jcm-11-01822-f002:**
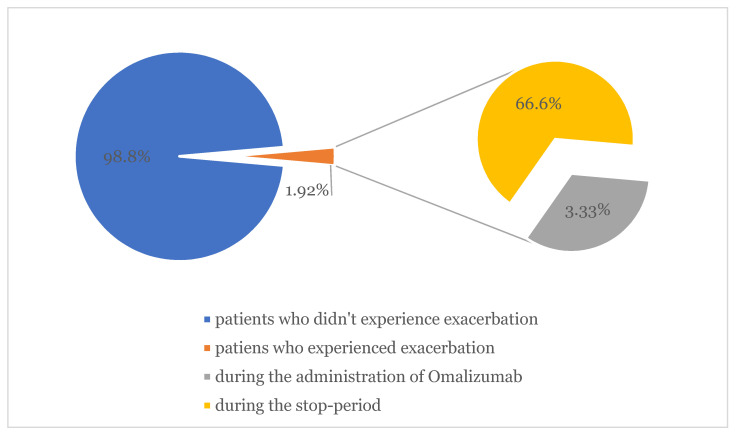
Exacerbation of CSU/CIU symptoms in the subgroup of patients who were taking omalizumab.

**Figure 3 jcm-11-01822-f003:**
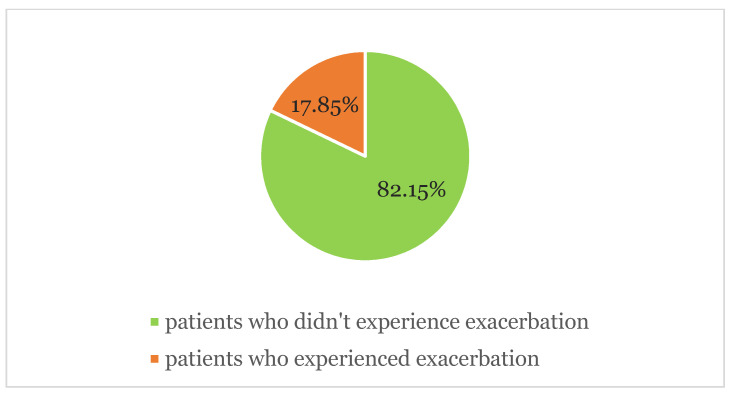
Exacerbation of CSU/CIU symptoms in the subgroup of patients who were taking anti-H1 therapy.

## Data Availability

Not applicable.

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
