# Peer review of "Effects of Vaccination against COVID-19 in Chronic Spontaneous and Inducible Urticaria (CSU/CIU) Patients: A Monocentric Study"

_jcm, 2022, doi:10.3390/jcm11071822_

Round 1

Reviewer 1 Report

Some problems remain with the presentation of Fig and Table and the way for subgrouping. Therefore, I present some questions and would like ask corrections to the manuscripts and charts.

  1. Since there are some reports of the initial onset and recurrence of urticaria after vaccination, it should be stated in the introduction.
  2. The first paragraph of the results shows the data of three subgroups. Because this data is important data in this paper, it should be presented using charts, etc. The total number of cases in the subgroup and the number of cases of urticaria exacerbation. Please make it visiblely.
  3. This Table 1 and Fig 1 do not reflect what is stated in the first paragraph. Also, the exacerbation of urticaria is an important message, but why is there no information on urticaria in Table 1?
  4. One of the subgroups has an 8-week washout group, and please explain why you are particular about the 8-week washout, including the discussion section. Omalizumab is usually given every 4 weeks, and we believe that some patients may have extended their withdrawal.

Author Response

Dar Reviewer, thank you for you advices. We list below our responses point by point

Some problems remain with the presentation of Fig and Table and the way for subgrouping. Therefore, I present some questions and would like ask corrections to the manuscripts and charts.

  1. Since there are some reports of the initial onset and recurrence of urticaria after vaccination, it should be stated in the introduction.

R: 1.    Dear reviewer thank you very much for your interesting suggestion. “Referred data suggest Urticaria and angioedema were the mainly observed Type-I reactions (stingeni 10.1111/all.14839), moreover cases of new onset urticaria have been reported by AA (10.7759/cureus.18102). An american registry -based study of 414 cases, firstly reported a percentage of  1.7 % of patients that developed cutaneous urticaria rash after Comirnaty and Moderna vaccines. 10.1016/j.jaad.2021.03.092“

  1. The first paragraph of the results shows the data of three subgroups. Because this data is important data in this paper, it should be presented using charts, etc. The total number of cases in the subgroup and the number of cases of urticaria exacerbation. Please make it visiblely.

R: Thank you, as requested we changed the Table and Figures, accordingly.

  1. This Table 1 and Fig 1 do not reflect what is stated in the first paragraph. Also, the exacerbation of urticaria is an important message, but why is there no information on urticaria in Table 1?

R: Dear reviewer your suggestion was really on-point, so we add to the table 1 the column concerning urticaria rash observed in our sample.

  1. One of the subgroups has an 8-week washout group, and please explain why you are particular about the 8-week washout, including the discussion section. Omalizumab is usually given every 4 weeks, and we believe that some patients may have extended their withdrawal.

R: Dear reviewer in Italy AIFA (Italian Medicines Agency) stated an 8-week stop period following every 6 months of Omalizumab treatment in CSU. Then we decide to specify it in the text.

Reviewer 2 Report

Thsi article faces the particular case of patients affected by pre-existing chronic spontaneous/inducible urticaria affording reactions to mrna vaccines for sars-cov-2. It is interesting and well writtne.

I have just some few comments:

Line 34: I think that it could be useful to clarify some clinical characteristics of chronic spontaneous urticaria; I mean to specify with few words the meaning of spontaeous.

Line 54-56: The authors suggest that potential markers have been suggested, but it is not clear if some of these markers have been selected for their characteristics in the daily use. I think it could be useful to specify. Or in alternative you could delete the sentence, because there is not referral to this topic in the data. 

Line 69: it seems that a stronger is referred to responses. But a is singular and responses is plural.

Line 89: 48.4 years

Line 106:  administered the same questionnaire, and not administered to the same...

Line 113-116: this is a ripetiotion of the previous period lines 105-111

Line 121: exacerbation of CSU sumptoms ....in which occasion? after....

Line 127: respectively is not necessary. I suggest to delete this word.

Author Response

Dear reviewer, thank you for your comments. and advices. We report below our changes point by point.

Thsi article faces the particular case of patients affected by pre-existing chronic spontaneous/inducible urticaria affording reactions to mrna vaccines for sars-cov-2. It is interesting and well writtne.

I have just some few comments:

-Line 34: I think that it could be useful to clarify some clinical characteristics of chronic spontaneous urticaria; I mean to specify with few words the meaning of spontaneous.

R: Dear reviewer, thank you for your accurately focusing on Chronic Spontaneous Urticaria. ....” is a relapsing mucocutaneous disease, characterized by hives (40%) angioedema (20%) or both (40%) in absence of apparent causative condition”.

- Line 54-56: The authors suggest that potential markers have been suggested, but it is not clear if some of these markers have been selected for their characteristics in the daily use. I think it could be useful to specify. Or in alternative you could delete the sentence because there is not referral to this topic in the data.

R:Thank you for giving us the opportunity to specify potential clinical or serological biomarkers, “such as D-dimer or total IgE level or coexistent autoimmunity conditions, that have been recently suggested and currently used in monitoring the disease”

- Line 69: it seems that a stronger is referred to responses. But a is singular and responses is plural.

R:Dear reviewer thank you, the term stronger is referred to response.

- Line 89: 48.4 years

Dear reviewer, we correct “of 99 females and 61 males (male/female ratio 1: 1.62), with a mean age of 48.4 years” as you suggested.

- Line 106:  administered the same questionnaire, and not administered to the same...

R: Dear reviewer, we delete “to” as you specified

- Line 113-116: this is a repetition of the previous period lines 105-111

Dear reviewer we suppress this period in agreement with your request “Every patient was administered the same questionnaire formulated in Italian. The questionnaire investigated the following: demographics including  age and sex; clinical features including time of onset of the CSU, personal history related to COVID-19 vaccination, including the status of CSU (relapsing, remitting or fully controlled) at time of vaccination.

-Line 121: exacerbation of CSU sumptoms ....in which occasion? after....

R: Dear reviewer thank you, we have added the administration of vaccine as you asked “The result of our investigation survey highlighted those 13 patients (8.12%), who experienced exacerbation of CSU symptomatology after vaccine administration.”

- Line 127: respectively is not necessary. I suggest deleting this word.

R: Dear reviewer thank you, we agreed to delete this word as you suggest “Exacerbation occurred in 9 patients (69.23%) after Pfizer mRNABNT162b2 (Comirnaty), in 4 patients (30.77%) after Moderna mRNA-1273 (Spikevax).

Round 2

Reviewer 1 Report

This paper is revised well according to my suggestion.